# Computational Approaches Drive Developments in Immune-Oncology Therapies for PD-1/PD-L1 Immune Checkpoint Inhibitors

**DOI:** 10.3390/ijms24065908

**Published:** 2023-03-21

**Authors:** Patrícia S. Sobral, Vanessa C. C. Luz, João M. G. C. F. Almeida, Paula A. Videira, Florbela Pereira

**Affiliations:** 1LAQV and REQUIMTE, Department of Chemistry, NOVA School of Science and Technology, Universidade NOVA de Lisboa, 2829-516 Caparica, Portugal; 2UCIBIO, Applied Molecular Biosciences Unit, Department of Life Sciences, NOVA School of Science and Technology, Universidade NOVA de Lisboa, 2829-516 Caparica, Portugal; 3Associate Laboratory i4HB—Institute for Health and Bioeconomy, NOVA School of Science and Technology, Universidade NOVA de Lisboa, 2829-516 Caparica, Portugal

**Keywords:** immune oncology therapies, immune checkpoint inhibitor (ICI), programmed cell death protein 1 (PD-1), programmed cell death ligand 1 (PD-L1), databases, web tools, computational methodologies, computer-aided drug design (CADD)

## Abstract

Computational approaches in immune-oncology therapies focus on using data-driven methods to identify potential immune targets and develop novel drug candidates. In particular, the search for PD-1/PD-L1 immune checkpoint inhibitors (ICIs) has enlivened the field, leveraging the use of cheminformatics and bioinformatics tools to analyze large datasets of molecules, gene expression and protein–protein interactions. Up to now, there is still an unmet clinical need for improved ICIs and reliable predictive biomarkers. In this review, we highlight the computational methodologies applied to discovering and developing PD-1/PD-L1 ICIs for improved cancer immunotherapies with a greater focus in the last five years. The use of computer-aided drug design structure- and ligand-based virtual screening processes, molecular docking, homology modeling and molecular dynamics simulations methodologies essential for successful drug discovery campaigns focusing on antibodies, peptides or small-molecule ICIs are addressed. A list of recent databases and web tools used in the context of cancer and immunotherapy has been compilated and made available, namely regarding a general scope, cancer and immunology. In summary, computational approaches have become valuable tools for discovering and developing ICIs. Despite significant progress, there is still a need for improved ICIs and biomarkers, and recent databases and web tools have been compiled to aid in this pursuit.

## 1. Introduction

Immune checkpoint inhibitors (ICIs) are a main breakthrough in immunotherapy due to their capacity to help the body’s immune system to recognize and attack cancer cells [1]. They work by blocking the pathways that allow tumors to evade detection and activation of the immune system. The immune checkpoints are expressed on the surface of immune cells and tumor cells and can either be upregulated or downregulated depending on the setting of the immune response and tissue environment. In cancer, when these checkpoints are upregulated, tumor cells can evade detection by the immune system, dampen their elimination by cytotoxic T cells and, subsequently, continue to grow [2]. The ICIs target these checkpoints and block their interaction expression, allowing the immune system to detect and fight the cancer cells.

As illustrated in Figure 1, immune oncology and, in particular, ICIs in cancer have become a markedly pursued subject of research in the last ten years.

The PD-1/PD-L1 immune checkpoint axis is crucial in regulating the immune response. Programmed cell death protein 1 (PD-1) is a receptor protein found on the surface of specific immune cells, including T-cells [3]. PD-1 binds to its ligand, programmed cell death ligand 1 (PD-L1), and triggers an inhibitory signal that halts T-cell activity against the target cell. PD-L1 is often overexpressed on the surface of cancer cells, and this is one way that cancer cells can evade destruction by the immune system [4].

Therefore, drugs that target the PD-1/PD-L1 axis can block this interaction and allow the immune cells to continue setting up a response against the cancer cells. These drugs are known as ICIs, and they have shown great promise in treating various cancers, including melanoma, lung cancer and bladder cancer. The present review aims to provide an overview of different computational approaches used to identify ICIs targeting the PD-1/PD-L1 axis or to ameliorate their efficacy or critical characteristics as therapeutics.

### 1.1. Methods

The literature used was indexed in Web of Science™ Core Collection, Current Contents Connect^®^, Derwent Innovations Index℠, KCI-Korean Journal Database™, MEDLINE^®^ and SciELO Citation before 18 January 2023. A total of 25,458 publications (13,901 articles, 91 proceedings, 2016 meeting abstracts, 671 editorials and 8517 reviews) were retrieved. This section describes the methods used to carry out the statistical analysis.

Data Collection: The data for this study were obtained by searching the aforementioned databases using the search terms “cancer” and “immune checkpoint inhibitor” prior to 18 January 2023. A total of 25,458 publications were identified and included in the analysis, comprising 13,901 articles, 91 proceedings, 2016 meeting abstracts, 671 editorials and 8517 reviews.

### 1.2. Publication Trends from 2013 to 2022 on ICIs

From 2013 until 2016 there was a steady increase in the number of publications related to ICIs per year, rising from 64 in 2013 to 900 during 2016. However, since 2016, we observed a notable increase in the number of reports. In the last five years, there was a nearly 94% increase in publications, with 6040 and 5984 publications in 2021 and 2022, respectively. The awarding of the 2018 Nobel Prize for Medicine in this area to James P. Allison and Tasuku Honjo may have contributed to the significant growth of interest in ICIs. The American immunologist James P. Allison contributed to the discovery of mechanisms underlying T-cell activation and was a pioneer in the development of immune checkpoint therapy for cancer, specifically using cytotoxic T-lymphocyte antigen-4 (CTLA-4) blockade. The Japanese immunologist Tasuku Honjo contributed to the discovery of mechanisms and proteins critical to the regulation of immune responses, which led to the development of novel immunotherapies against cancer, namely the development of anti-programmed cell death protein 1 (PD-1) cancer immunotherapies.

Worldwide, more than three-quarters of the publications (i.e., 21,372 out of 25,400, corresponding to 84%) since 2013 have been linked to five countries: USA, China, Japan, Italy and France. The most productive country in this field has been the USA, contributing 9486 outputs across this period (37%). However, it is necessary to consider that the publications may include researchers from several countries.

Over 2000 journals have been selected as containing publications relating to ICIs in cancer since 2013. The top twenty-five journals reporting the above theme are listed in Figure 2.

The four most-published journals are all open access, namely *Cancers* from MDPI, *Frontiers in Immunology* and *Frontiers in Oncology* from Frontiers and *Journal for Immunotherapy of Cancer* from Society for Immunotherapy of Cancer (SITC) (Figure 2), which also contributed to the wide dissemination of this subject. The first generalist journal to appear is the *International Journal of Molecular Sciences* (IJMS) of the MDPI publisher; in positions 23rd and 24th are *Frontiers in Pharmacology* (Frontiers publisher) and *Scientific Reports* (Nature publisher), respectively (see Figure 2). Although we perceive that, in general, older publications accumulate more citations, and the publications analyzed in this theme are mostly very recent (representing 94% in the period of 2017–2022), there are clearly two journals that stand out with numbers of citations per publication of 81 and 57 for *Clinical Cancer Research* and *Cancer Immunology Research*, respectively (see Figure 2). Interestingly, these two most-cited journals are both from the American Association for Cancer Research publisher.

### 1.3. From Antibodies to Small Molecules: The Advancements in PD-1/PD-L1 ICIs

In 2011, cancer treatment was revolutionized with the first approval of an ICI by the U.S. Food and Drug Administration (FDA), Ipilimumab [5], establishing a new field of research–immuno-oncology. Ipilimumab is a monoclonal antibody (mAb) against the immune checkpoint CTLA-4 developed by the Nobel Prize winner James P. Allison. Regarding the PD-1–PD-L1 axis, the first anti-PD-1 Ab, pembrolizumab, developed following the findings of Nobel Prize winner Tasuku Honjo, was approved by the FDA in 2014 to treat advanced melanoma. Two years later, in 2016, the first anti-PD-L1 Ab, atezolizumab, was approved for the treatment of bladder cancer. To date, the FDA has approved seven ICIs mAbs for the PD-1/PD-L1 pathway: four anti-PD-1 (nivolumab, pembrolizumab, cemiplimab and dostarlimab) and three anti-PD-L1 (atezolizumab, avelumab and durvalumab) [6,7]. Small molecule ICIs of the PD-1/PD-L1 interaction have been explored to obviate the drawbacks of mAbs such as lack of oral bioavailability, prolonged retention in the tissues and poor permeability [7]. Although no small molecules have yet been approved as PD-1/PD-L1 ICIs, eight small molecules are already in clinical trials, mostly in the early stages [8], as shown in Table 1. The small molecules navtemadlin (**3**) and ciforadenant (**5**), shown in Table 1, are also being tested in combination clinical trials with mAbs avelumab and atezolizumab [8]. In addition to these two small molecules, there are ten more small molecules (**9**–**18**) being used in combination clinical trials with mAbs, as shown in Figure 3.

The development of a drug from concept to market currently takes 13–15 years [9] with an average investment of $2–3 billion [9]. To reduce the length and cost of the drug discovery and development process, drug designers have been exploring computer-aided drug design (CADD) tools [10,11,12,13,14] over the past few decades to design and identify new drug-like chemical entities that can overcome clinical attrition and lack of efficiency.

In this review, we highlight the computational methodologies applied to the discovery and development of PD-1/PD-L1 ICIs for improved cancer immunotherapies, with greater focus in the last five years. The use of CADD structure- and ligand-based virtual screening processes, molecular docking, homology modeling and molecular dynamics (MD) simulations methodologies essential for successful drug discovery campaigns focusing on small molecule ICIs will be addressed. Databases and web tools in the context of cancer and immunotherapy will be also reported in this review and in the Appendix A. In this review, the most relevant current computational methodologies for binding mode and binding affinity prediction of the PD-1/PD-L1 immune checkpoint are highlighted.

## 2. Databases and Web Tools in the Context of Cancer and Immunotherapy

When developing immunotherapies for cancer, databases are a valuable resource that provide researchers and clinicians with access to a wealth of information about cancer cells’ genetic and molecular characteristics and the immune system. These databases collect and store a vast amount of data from various sources, such as clinical trials, genomic sequencing and gene expression profiling.

The most-used databases are the global repositories for nucleotide sequences of the International Nucleotide Database Consortium [15,16,17], as they harbor almost all the relevant sequences known until now, excluding the databases of reads that are not individually annotated. Even if they present different names, addresses or user interfaces, they share the very same body of data regarding the sequences. These databases are usually served by a specific database for translated coding sequences (CDS) and their annotation. Apart from the NCBI Peptide [15], the most popular database is UniProtKB from the European Molecular Biology Laboratory (EMBL) [18]. The latter cooperates closely with InterPro [19] regarding the functional annotation and provides a fully integrated view of the knowledge of a given sequence, especially regarding entries for its Swiss-Prot division. It may be the ideal entry point for obtaining annotation for a given protein or the underlying family/domain architecture.

Functional annotation is, nowadays, enabled by a set of different and concurrent databases with their specific models. Being different in nature and source sequence sets, the concurrency of matches imparts significance, and probably detail, to the annotation of a novel sequence. In this field, InterPro (and InterProScan) shines due to its meta-database structure, the models (“signatures”) belong to the federated partners [20,21,22] and the InterPro matches are accompanied by a consolidated annotation that makes it the resource of choice for functional annotation. Regarding proteins and their role in the cell, structural information is of paramount importance, even if it is not always known. Nowadays, there is an ever-increasing volume of available structures, and they are mostly stored at Protein Data Bank (PDB) [23]; this database comprises both experimentally derived structures and a growing amount of results from homology modeling.

Databases of annotated genes comprise the next level of information consolidation, as they connect the individual sequences, plus their annotation to the genomic assembly and the genomic variability or even gene expression. The largest database of this scope by far is Gene from the National Center for Biotechnology Information (NCBI), and it connects almost seamlessly with Genome, the NCBI database for genomic assemblies [15]. Its entries are also present, through the all-pervasive body of cross references, in other databases such as the University of California, Santa Cruz, Genomes (UCSC) [24]. Individual genes are of limited use when dealing with the physiological conditions of the cell or other high-level phenomena to address the requirements of dealing with huge pools of genes, and their expression ontologies were developed into group genes according to their function and process and the localization of the product. The Gene Ontology (GO) project has been the main reference in this role [25], and there are derivatives and subsets, even if the rationale remains. Other classification approaches may be used, such as the one used by Database for Annotation, Visualization and Integrated Discovery (DAVID) [26], using functional annotation on large sets of genes. Other ways of classification are possible, and available, based on increased levels of data integration, such as the Molecular Signatures Database (MSigDB) [27], the PRO [28] or the Human Protein Atlas [29].

Regarding the gene expression study results, the Gene Expression Omnibus (GEO) database [30] is the main destiny for experimental data. The deposited information, and new datasets gained through insight when they are crossed with known interactions, are available through STRING [31] or the even more detailed content of GeneMANIA [32]. In any case, the accepted reference for mapping a given set of genes to a pathway has been, for a long time, the KEGG Pathway database [33]. The Online Catalog of Human Genes and Genetic Disorders (OMIM) bridges the gap between the genotype and the phenotype while focusing on the field of human mendelian disorders [34].

Departing from the general point of view and looking into cancer situations, there are a multitude of avenues and resources, depending on the required scale. Mutations and their putative outcomes can be inferred with the help of both divisions of the FannsDB [35,36]. Looking into the cell and their mutation repertoire in an integrated fashion, databases such as the Broad’s Cancer Cell Line Encyclopedia [37], the Cancer Genome Atlas [38];,the cBioPortal [39], TIMER [40], GTEx portal [41] and GEPIA [42] may be relevant stops for finding expression data analysis, pharmacologic impact characterization or epigenetic information. Those projects incorporate and analyze an ever-increasing number of results according to stable pipelines that highlight specific aspects of the data.

Other relevant sources of database stored data are dedicated portals that centralize different sources and facilitate in different degrees the processes of querying and data retrieval. The Genomic Data Commons (GDC) of the National Cancer Institute [43] or the Broad’s GDAC—Firehose are good examples for the kind of facilities alluded. The ability to associate medical imaging results with clinical and molecular data is well represented by the Cancer Digital Slide Archive [44]; in this case, the connection with the Cancer Genome Atlas is established through the images’ meta-data.

The interactions between tumors, the immune system and pharmacology are addressed in portals such the Tumor Immune System Interactions and Drug Bank (TISIDB) [45]. Regarding the work on ICIs, the availability of T-cell receptor-related databases is important. Databases such as VDJDB [46] or PIRD [47] are complemented by much smaller but manually curated resources such as McPAS-TCR [48] or TBAdb [47], where the latter group trades quantity for quality. Other databases are more ambitious in their scope, integrating a more diverse dataset. This is the case for the Human Primary Cell Atlas [49] and The international ImMunoGeneTics information system^®^ [50].

Importantly, databases serve as essential resources for identifying potential drug targets and understanding the molecular mechanisms of drug action. Several critical databases in this context are described in Appendix A.

## 3. Computational Methods for Predicting Checkpoint Inhibitors

### 3.1. Small Molecules

Small molecule-based immunotherapies, compared to peptide- and especially Ab-based therapies, offer advantages such as a higher oral bioavailability, overall lower costs and a relative shorter half-life, which leads to less potential toxicity and adverse effects [51]. However, despite the pharmacokinetic benefits, small molecules have been receiving limited attention, probably due to the intrinsic difficulty of mimicking conventionally complex protein–protein interactions (PPIs) [52]. It was only recently that small molecules started receiving attention, taking on a more relevant role in immune checkpoint therapies, especially as inhibitors of the PD-1/PD-L1 interaction. This late and complex investigation might be due to the PD-1/PD-L1 large binding interface, spanning ~1970 Å^2^, devoid of well-defined binding pockets [53]. The first small molecules with a promising inhibitory activity against PD-L1, a series of compounds containing a biphenyl group, were disclosed by Bristol-Myers Squibb (BMS) in 2015 [54]. However, the binding mechanism was found by Holak’s group, who discovered that the BMS compounds induced the dimerization of the PD-L1 protein. The unveiling of two co-crystal structures of PD-L1 in complex with small molecule inhibitors, BMS-202 (**19**) and BMS-200 (**20**) (PDB ID: 5N2F and PDB ID: 5J89, respectively), brought light to structure-based drug design [55,56]. Mittal et al. [57] applied molecular docking studies to a PD-1/PD-L1 complex with BMS inhibitors against the PD-L1 protein and discovered that the residues Tyr56, Asp122 and Lys124 play crucial roles in ligand binding to the PD-L1 protein (Figure 4), an important piece of information for the design of inhibitors of PD-L1. These findings also led to the development of different molecules containing a biaryl moiety, following a ligand-based approach. 

Qin et al. discovered and explored the structure activity relationships (SARs) of a series of [1,2,4]triazolo[4,3-a]pyridines scaffold inhibitors, showing that a ring fusion strategy could be employed for designing analogues of the Bristol-Myers Squibb chemical compounds such as BMS-202 (**19**) and BMS-200 (**20**), as shown in Figure 5 [59]. In the following work, they also developed and studied SARs of indoline scaffold compounds [60], and, afterwards, 4-arylindolines containing a thiazole moiety [61], with one of them, A30 (**21**) in Figure 5, reaching an IC_50_ value of 11.2 nM. This showed that compounds bearing a thiazole moiety exhibited significantly increased activity compared to compounds containing a thiophene group. The authors applied molecular docking to confirm the binding mode of A30 (**21**) to PD-L1 binding sites using the BMS-200/PD-L1 complex as a template, showing that A30 (**21**) fit well within the inhibitor binding pocket [61]. Cheng et al. [62] chose a combined structure and ligand-based approach and started with docking-based virtual screening. The hit compound was then improved through SAR studies, and a series of resorcinol dibenzyl ethers were synthetized, with compound NP19 (**22**) in Figure 5 inhibiting the PD-1/PD-L1 interaction with an IC_50_ value of 12.5 nM in Homogeneous Time-Resolved Fluorescence (HTRF) binding assays. Also based on the pharmacophoric model of BMS compounds, and with the use of molecular docking as a corroborative technique [63,64,65,66,67,68,69,70,71,72,73], Dai and co-workers [74] obtained 1-methyl-1H-pyrazolo [4,3-b] pyridine derivatives, with the best compound, D38 (**23**) (Figure 5), having an IC_50_ value of 9.6 nM. Liu et al. [75] reported benzo[c][1,2,5]oxadiazole derivatives, with molecule L7 (**24**) in Figure 5 exhibiting an IC_50_ value of 1.8 nM. Wang et al. [76] designed biphenyl pyridine derivatives, with the most potent compound (**25**) showing an IC_50_ value of 3.8 nM [59].

In 2022, DiFrancesco and co-workers [51] began a high-throughput virtual screening of approximately 3.7 million lead-like molecules from the ZINC online repository using a rigid-receptor docking approach against both human PD-1 and PD-L1. This first step showed possible small molecule tractability of only the PD-1 binding. This lead-like dataset followed, as criteria, a molecular weight between ≥250 and ≤350 g/mol, an XlogP value ≤3.5 and a number of rotatable bonds ≤7. The next step involved molecular docking of the National Cancer Institute (NCI) compound dataset only against the PD-1 binding pocket. The top ~0.1–5% of molecules obtained from both datasets were clustered into separate bins using the ChemMine server binning tools. Considering the cost, commercial availability and binding to key residues of PD-1 and PD-L1, 40 top hit compounds were selected. The dynamic binding performance of these compounds was further investigated using long 100 ns MD simulation with the Desmond Molecular Dynamics package. Despite its low molecular weight (415.5 g/mol), compound NSC631535 (**26**) in Figure 6 reveals itself to be a potentially stable binder at the PD-1 interface pocket, exhibiting an experimental IC_50_ of 15 µM.

Chandrasekaran et al. [77] used a drug repurposing approach consisting of e-Pharmacophore modeling, molecular docking and then MD simulation. All the in silico preparations, such as protein and ligand preparation, molecular docking and MD, were performed using the Maestro 12.6 modeling package. The three-dimensional crystal structure of human PD-L1 (PDB ID: 5J89) co-crystallised with a small molecule inhibitor, BMS-202 (**19**), was retrieved from the PDB. The treated PD-L1 protein was docked with INCB086550 (**5**) in Table 1, an Incyte corporation hit molecule for PD-L1 inhibition. Based on the observed interactions, an e-Pharmacophore hypothesis with six crucial characteristics was determined, comprising two acceptors of hydrogen bonds, one donor of hydrogen bonds, one positively ionizable group and two aromatic rings. To reveal FDA-approved candidates that can be repurposed against PD-L1, the pharmacophore characteristics were introduced into the PHASE module with a matching threshold of four out of six features. The screening retrieved 324 FDA-approved drugs with a fitness score of ≥1. The ten top hits were then compared with a clinical trial candidate, IN-35 (**27**) in Figure 6. Compounds with a fitness score greater than one were selected for extra-precise flexible docking at the PD-L1 dimer interface. Mirabegron (**28**) and indacaterol (**29**), shown in Figure 6, exhibited docking scores against the PD-L1 dimer (PDB: 5J89) of −9.213 kcal/mol and −8.023 kcal/mol, respectively. The binding-free energy of mirabegron (**28**) and indacaterol (**29**) was determined by MM-GBSA analyses, suggesting that mirabegron (**28**) uses less energy to create a more stable complex. MD simulation confirmed that the mirabegron (**28**) complex demonstrated a similar pattern of deviation in correlation with IN-35 (**27**), maintaining the interaction with the active key amino acids during the simulation period. Fattakhova and co-workers [78] reported a drug repurposing strategy, using a structure and ligand-based screening of ZINC15 database, that includes ~10,000 approved and investigational drugs. The AutoDock Vina docking algorithm was used to carry out structure-based docking of the drug molecules to all PD-L1 dimer interfaces (PDB IDs: 5N2F, 5NIU, 6R3K, 5J89, 5J8O, 5N2D, 6NM8). The top 1000 molecules with the best docking scores were selected. The ligand-based virtual screening of ZINC15 was performed using ROCS 3.4.1.0 (Open Eye Scientific Software, Santa Fe, NM. http://www.eye-sopen.com (accessed on 28 February 2023)), a database that aligns and ranks drugs based on the similarity to a given 3D structure. The compounds with a higher similarity, as indicated by higher Tanimoto Combo scores across the seven crystal ligands, were then combined with the previously obtained 1000 molecules. These top ROCS hits were then further subjected to docking against the high-resolution PD-L1 crystal structure (PDB: 5N2F). After studying MD and performing HTRF binding assays, pyrvinium (**30**) in Figure 6, an FDA-approved anthelmintic drug, showed the highest activity, with an IC_50_ value of approximately 29.66 μM. Since pyrvinium (**30**) is an approved drug, it may be a good starting point for the design of even more potent analogues as PD-1/PD-L1 inhibitors.

Similarly, Urban et al. [79] also employed a pharmacophore-based virtual screening (PBVS) approach. The structure-based pharmacophore model was obtained with the crystal structures of the PD-L1 dimer in complex with BMS-8 (PDB ID 5J89), BMS-202 (PDB ID 5J8O) in Figure 5, BMS-1001 (PDB ID 5NIU) and BMS-1166 (PDB ID 6R3K) by Pharmit server software. PBVS was performed in the PubChem database, which contains over 90 million chemical structures, and the matching compounds were docked to the PD-L1 dimer using the AutoDock Vina program. The PD-L1/inhibitor complexes were subjected to energy minimization using the Smina minimization feature of the same program, and complexes with energies lower than 8 kcal/mol were selected for further screening. Screening was performed using pharmaceutically relevant descriptors and properties of lead-like drugs. The first screening was performed using the QikProp program and the second after docking optimization by the SwissADME web tool. After molecular docking screening using the Glide module of the Schrödinger Suite, the MD simulations showed that nine out of ten compounds established stable complexes with the PD-L1 dimer, as shown by the analysis of MD trajectories. Additionally, molecular mechanics Poisson–Boltzmann surface area (MM-PBSA) calculations revealed low binding energies for the interactions of these ligands with the PD-L1 dimer, indicating that the identified molecules could be used for the design of novel inhibitors of PD-1/PD-L1 interactions.

Luo and co-workers [80] also used a structure-based PBVS approach, applying Discovery Studio 4.5 to establish a pharmacophore model using PD-L1/small molecule inhibitors downloaded from PDB after screening marine small molecule databases such as the Comprehensive Marine Natural Products Database (CMNPD, https://www.cmnpd.org/ (accessed on 28 February 2023)) and the Seaweed Metabolite Database (SWMD, http://www.swmd.co.in/ (accessed on 28 February 2023)) according to the properties of the pharmacophore. The compounds matched from the different databases were transferred to a list for another virtual screening based on pharmacophore characteristics, yielding a total of 12 hits. Then, two compounds were selected for further evaluation based on their molecular docking scores. Afterward, ADME (absorption, distribution, metabolism and excretion), toxicity and docking studies were performed using the SwissADME, ProTox-II and CDOCKER programs, respectively. Molecule 51320 (**31**) in Figure 6 was selected for MD analysis using GROMACS. Moreover, molecule 51320 (**31**) maintained a steady conformation with PD-L1, indicating that the compound could be used or improved to potentially become an inhibitor of PD-L1.

On the other hand, Kumar and co-workers [81] began with a virtual screening of 32,552 compounds from the Natural Product Atlas database for PD-L1 inhibitors. First, all ligands were filtered using ADME and drug-likeness criteria using the QikProp tool, and then the filtered ligands were virtually screened via three subsequent steps, including (i) high-throughput virtual screening (HTVS), (ii) standard precision (SP) screening and (iii) extra precision (XP) screening protocols. Five natural compounds, namely neoenactin B1 (**31**), actinofuranone I (**33**), cosmosporin (**34**), ganocapenoid A (**35**) and 3-[3-hydroxy-4-(3-methylbut-2-enyl)-2-methylidene-cyclohexanone (**36**), shown in Figure 6, were collected, and the joint results of binding free energy calculation and MD simulation support the use and further development of these compounds as PD-L1 ICIs in cancer immunotherapy.

With a study based on molecular docking using GOLD software (GOLD 5.3 release, Cambridge Crystallographic Data Center, Cambridge, UK), Vergoten et al. [82] showed the potential of pseudoguaianolide sesquiterpene lactones, especially BRT (**37**), to interact with PD-L1 homo and dimer forms. Besides BRT (**37**), 15 other pseudoguaianolide sesquiterpene lactone analogues to BRT (**37**), including helenalin (**38**), gaillardin (**39**), bigelovin (**40**) and coronopilin (**41**), as shown in Figure 6, were docked to PD-L1 homo and dimer forms. BRT (**37**) seemed to form a more stable complex with the dimeric form, with a calculated empirical energy of interaction (ΔE) value of −63.1 kcal/mol, similar to the ΔE of PD-L1/BMS-202 (**19)**, shown in Figure 5 (−73.4 kcal/mol) under the same conditions. The docking analysis of chamissonolide (**42**), shown in Figure 6, also yielded good results, with a ΔE of −64.8 kcal/mol, providing a direction for the design of novel PD-L1 binders incorporating an SL-like tricyclic scaffold. The structure of the ligands was optimized before docking using a classical Monte Carlo conformational searching procedure, as established in the BOSS software [83].

#### Lessons from Small Molecules Targeting Other Immune Checkpoints

Although the most recent studies involving cancer immunotherapy and computer methods mainly target PD-1/PD-L1 immune checkpoints, computer approaches have also been useful for discovering novel small molecules for other immune checkpoints. Qayed and co-workers [84] reported a combination of ligand- and structure-based modeling studies, designing and synthetizing novel hybrids of 2-indolinone-thiazolidinone, a well-known scaffold with anticancer activity. Twenty-six hybrids were tested in vitro for CDK2 inhibition, a key checkpoint enzyme, and compared to the reference enzyme inhibitors, such as sunitinib (**43**), nintedanib (**44**) and semaxanib (**45**) (Figure 7). To support their biological activities, broad docking and MD simulations were performed. In addition to the confirmed consistency between the in vitro and in silico studies, ADME and Tox studies using SwissADME and pkCSM also predicted good pharmacokinetic properties and low toxicity.

CD73, a glycosylphosphatidylinositol (GPI)-linked cell surface enzyme, has been receiving increasing attention in the past decade. Although some inhibitors have been identified through in vitro assays, none of them have passed preliminary clinical trials. Lyu et al. [85] aimed to identify novel inhibitors by using a structure-based virtual screening approach, retrieving about 500 molecules with a high binding affinity from the Chemdiv-Plus database. These molecules were filtered by drug properties and pharmacokinetics using the Stardrop program (Release 6.5.0, Optibrium Ltd., Cambridge, UK), resulting in 68 small molecules. Phelligrin-based compounds showed the best experimental inhibition activities and may serve as lead compounds for the development of better analogues with the insight provided by the docking studies.

### 3.2. Peptides

In addition to small molecules, computational methods also played a crucial role in the discovery of novel peptides as immunotherapeutic drugs.

Li et al. [86] designed several PD-1 binding peptides using a computational de novo design method, which was divided into four steps: (1) identification of key anchor residues and building of a scaffold library; (2) screening of the scaffold to identify fragments in which the identified anchor residues are present; (3) transfer of the fragments onto the scaffold; and finally (4) sequence design and structure optimization using the Rosetta modeling package (https://www.rosettacommons.org/software (accessed on 28 February 2023)). The best peptides were then selected for in vitro studies.

The scaffold library was built by extracting scaffold fragments from 22,912 protein crystal structures in the PDB. The key PD-L1 anchor residues, Tyr56, Arg113, Ala121, Asp122 and Tyr123, were identified using three prediction tools, Robetta, KFC2 and PredHS, with the PD-1/PD-L1 complex (PDB ID 4ZQK) as the input structure. Scaffold screening was performed by superposing each fragment from the library onto the anchor residues using the Cealign algorithm in the Pymol program. The scaffold fragments with a root-mean-square deviation (RMSD) inferior to 2.0 Å were kept for the next step. Since one scaffold was unable to accommodate all five key residues due to their relative positions and structural properties, two scaffold fragments were selected. To transform the two scaffolds into a continuous peptide, Rosetta module Kinematic loop modeling was applied, and the optimization of the sequence design and structure minimization was performed employing the Backrub module. Finally, the resulting peptides in complex with PD-1 were enhanced using Rosetta’s Relax module and scored using the InterfaceAnalyzer module of Rosetta. The most potent peptide, Ar5Y_4 (**46**), as seen in Figure 8, showed a KD value of 1.38 ± 0.39 μM, comparable to the binding affinity of the cognate PD-L1.

On the other hand, Manavalan and co-workers [87] created a web server, PIP-EL (www.thegleelab.org/PIP-EL (accessed on 28 February 2023)), a free access automated computational method that allows a faster identification of novel Proinflammatory Inducing Peptides (PIPs) using the strategy of ensemble learning (EL). PIPs have the ability to induce proinflammatory cytokines and have been used as an antitumor agent, an antibacterial agent and a vaccine in immunotherapies.

To build the dataset, experimental validated positive and negative peptides were extracted from IEDB (https://www.iedb.org/ (accessed on 28 February 2023)). A peptide was considered positive if it induced any one of the proinflammatory cytokines (i.e., IL1α, IL1β, TNFα, IL6, IL8, IL12, IL17, IL18 and IL23) in T-cell assays of human and mouse. Since the dataset was imbalanced, a random under-sampling technique to generate 10 balanced models for each composition was applied, leading to the combination of 50 random forest models. PIP-EL reached a Matthews’ correlation coefficient (MCC) of 0.435 in a 5-fold cross-validation test, which is ~2–5% higher than that of the individual classifiers. Additionally, the performance of PIP-EL was tested on an independent dataset, resulting in an MCC of 0.454. Therefore, PIP-EL can be a useful tool for researchers working in the field of peptide therapeutics and immunotherapy.

Another web server, PDL1Binder (http://i.uestc.edu.cn/pdl1binder/cgi-bin/PDL1Binder.pl (accessed on 28 February 2023)), was developed by He and co-workers [88]. The dataset is composed of 80 PD-L1 binding peptides obtained by next-generation phage display (NGPD) and is used for the development of computational models for identifying PD-L1 binding peptides. Instead of random forest-based models, as in the previous work, a Support Vector Machine-based model composed of four different peptide descriptors and an optimal feature selection approach chosen from five feature selection strategies outperformed the models developed with eleven other machine learning (ML) algorithms. This work offers promising PD-L1 binding peptide candidates for further investigations.

Antunes et al. [89] also implemented a new platform using Jupyter Notebook and the Docker program called HLA-Arena, which allows the analysis and even structural modeling of peptide–HLA complexes. This platform enables the use of computational tools such as APE-Gen, DINC, MODELLER and MHCflurry. This environment can be used to perform structural analyses for personalized cancer immunotherapy, neoantigen discovery or vaccine development.

Li and co-workers [90] described an in silico virtual screening and biomolecular interaction analysis (BIAcore) against an internal peptide library to discover novel PD-1/PD-L1 inhibitor peptides. Screening was performed using molecular docking and RRQWFW-NH_2_ (**47**) and RRWWRR-NH_2_ (**48**), which were found to have the best docking scores, as shown in Figure 9. Next, MD simulations and surface plasmon resonance (SPR) were used to further corroborate their predictions. The in vitro studies were consistent with the computational data, providing some insight on new candidate peptide drugs for cancer immunotherapy.

On the other hand, Bojko and co-workers [91] designed and synthetized potential blockers/inhibitors of the PD-1/PD-L1 pathway. The sequences of the peptides were based on the binding sites of PD-1 to PD-L1, as shown by the crystal structure of the complex, and also on molecular mechanics with generalized Born and surface area solvation (MM/GBSA). The computational results were confirmed by in vitro and cell assays, such as Surface Plasmon Resonance (SPR), which studies the interactions, and cell assays that study their inhibitory activity.

PD-1(122–138) C123-S137C (**49**) in Figure 10 has been shown to have the best inhibitory activity, so its 3D NMR structure was established and the binding site to PD-L1 was determined using molecular modeling methods. These results showed that the peptides synthetized are able to mimic the PD-1 interactions and block the PD-1/PD-L1 interaction.

One of the most recently developed computational methods, AlphaFold, has revolutionized structural biology by predicting highly accurate structures of proteins and their complexes with peptides, antibodies and proteins [92,93,94]. In addition to that, AlphaFold can also be useful for protein–peptide systems and drug discovery, as it can help identify the highest affinity binder among a set of peptides. Chang and Perez [92] presented a new competitive binding assay using AlphaFold to predict the structures of the receptor in the presence of peptides. The authors tested the application on six protein receptors for which they possessed the experimental binding affinities to several peptides and obtained the predicted structures (bound form) for the higher affinity peptides [92]. They concluded that this assay had a better prediction score for identifying stronger peptide binders that also adopt stable secondary structures upon binding [92]. Johansson-Åkhe et al. [93] also show the importance of AlphaFold, in this case, AlphaFold-Multimer. AlphaFold-Multimer is shown to be capable of predicting the structure of peptide–protein complexes with acceptable or better accuracy than previous models and also has the capacity to predict whether a peptide and a protein will interact, thus improving peptide–protein docking [93].

### 3.3. Antibodies (Abs)

The recent advances on computational approaches to improve immunotherapies for cancer are mostly based on the discovery of novel small molecule and peptide/peptidomimetic inhibitors of immune checkpoints [95,96]. On the other hand, in the area of Abs, recent investigations have had a greater focus on the discovery and better elucidation of the binding modes of Abs onto antigens that can guide the development of variants with higher affinity and specificity. The computational approaches may also improve pharmacokinetic properties of mAbs, which tend to be worse compared to peptides and small molecules.

On the other hand, the study of mechanisms and binding modes, especially of already approved Abs for the treatment of cancer, can help elucidate the mechanisms of smaller molecules and even lead to the discovery of potential novel molecules.

In this context, Tavares et al. [97,98] investigated in silico, by employing quantum chemistry methods based on the Density Functional Theory (DFT), the binding energy features of the receptor PD-1 in complex with pembrolizumab, an FDA-approved Ab for advanced melanoma (see Figure 11). Their computational findings allowed for a greater knowledge of the binding mechanisms, helping in the development of better Ab-based drugs or even small molecules.

Wen et al. [99] also studied the binding mechanism of Abs against PD-L1. They used the computational alanine scanning method (MM/GB/ASIE method) to calculate residue-specific binding energy contributions for five systems: PD-L1/KN035 (Figure 12), PD-L1/atezolizumab, PD-L1/avelumab, PD-L1/durvalumab and PD-L1/BMS-936559 (PDB IDs: 5JDS, 5XXM, 5GRJ, 5X8M and 5GGT, respectively). Their hotspots were predicted, and their results showed that PD-L1 Met115 and PD-L1 Tyr123 are important hotspots for all five complexes. In addition, it also showed that the crucial mAbs residues binding to PD-L1 Met115 and PD-L1 Tyr123 are very similar to each other. The experimental measurements available for KN035 and atezolizumab are in agreement with the computational results, with a correlation coefficient of 0.87 for PD-L1/KN035 and 0.6 for PD-L1/atezolizumab. The computational results also found more interactions on PD-L1/KN035, which is also in agreement with the experimental results.

In terms of Ab–antigen interactions, Myung and co-workers [100] developed a free access web server named CSM-AB (http://biosig.unimelb.edu.au/csm_ab/datasets (accessed on 28 February 2023)), an ML method with the capacity to predict binding affinity. This is made possible by considering structural features such as graph-based signatures and atomic interactions of Ab–antigen interface residues. CSM-AB achieved a Person’s correlation of 0.64 on blind tests and also works as a docking score function, accurately ranking poses.

## 4. Binding Mode and Binding Affinity Prediction of the PD-1/PD-L1 Immune Checkpoint

### 4.1. Target Predictions of Small Molecules and Peptides

Several small molecules targeting the PD-1/PD-L1 interaction have been identified in the last years, some of them comprising moieties such as urea (e.g., CA-170 (**1**)), guanidine (e.g., abemaciclib (**4**), ciforadenant (**6**), 6-thio-2′-deoxyguanosine (**10**), SF1126 (**15**)), sulfone (e.g., navtemadlin (**3**), vismodegib (**11**)), biphenyl (e.g., INCB086550 (**5**)) and *N*-heterocycle (e.g., tomivosertib (**2**), pexidantinib (**9**), carbozanitinib (**12**), cediranib and olaparib (**13**), itacitinib (**14**), sitravatinib (**16**), talazopanib (**17**), apatinib (**18**)), which are already in the clinical phases (Table 1 and Figure 3). However, the clinical results of targeting multiple immune-related targets still need to be elucidated; therefore, some works have recently been reported to identify small molecule targets using computational approaches in this regard [101,102,103]. Zhang et al. [101] reported using the PharmMapper database [104] to identify the targets of the small molecule lomustine (a nitrosourea derivative, **50**), which is investigated for the treatment of primary glioblastoma (see Figure 13). Pan et al. [102] used the BATMAN-TCM web tool [105] to identify 288 targets and 53 bioactive compounds from Qingfei Jiedu decoction (QFJDD, an empirical Chinese prescription prepared from several herbals) that regulates PD-L1 expression in lung adenocarcinoma (LUAD). In the end, six flavonoid derivatives, including quercetin (**51**), luteolin (**52**), kaempferol (**53**), wogonin (**54**), baicalein (**55**) and acacetin (**56**), and 22 hub genes were identified (Figure 13). Huang et al. [103] reported the use of PASS web resources for the validation of MYC/CXCL8 (C-X-C motif chemokine ligand 8)/TIMP1 (TIMP metallopeptidase inhibitor 1) oncogenes, which regulate immune response in an antitumor direction by mediating PD-L1, as potential drug targets for RV59 (**57**), an anthraquinone derivative with anticancer activity against NCI human colon cancer cell lines (Figure 13). Mittal et al. [57] developed an in silico structure-driven approach to screen and classify small molecules from the Asinex Signature library for the investigation of the inhibition of the PPIs between PD-L1 and PD-1/CD80 and its overexpression on cancer cells. In the end, two molecules, H5 (**58**) and H12 (**59**), were proposed as ICIs (Figure 13). The intersection between the two predicted targets, lomustine (**50**) and glioblastoma-related targets, was 59 active targets [101]. PPI, gene enrichment analysis, gene difference analysis, molecular docking and MD simulation have been used by the authors to screen out three effective targets of lomustine (**50**): HMOX1 (heme oxygenase 1), AKT1(AKT serine/threonine kinase 1) and EGFR (epidermal growth factor receptor) [101]. The same target, EGFR, was proposed by a different approach for QFJDD bioactive compounds (**51**–**56**) through a network construction and analysis comprising PPI, GO, KEGG and DAVID pathway analyses and molecular docking [102]. Several selected targets were associated with numerous human cancers, e.g., AKT1 [101], EGFR [101,102], HIF-1 (hypoxia inducible factor 1 subunit alpha) [102], MYC [103] and CXCL8 [103]. Furthermore, some of the selected targets seem to be related to an inflammatory response (CXCL8 [103]) or responses to viral infection (JUN [102], NFκB [102], and CD80 [57]).

Molecular docking simulations were used by the three works [101,102,103] to evaluate the potential interactions of the small molecule ICIs with the proposed targets. For example, in the molecular docking of lomustine (**50**) [101] and quercetin (**51**) [102] with an EGFR target, hydrogen bonds between lomustine (**50**) and Asp855 and Thr854 residues [101] and between quercetin (**51**) and Lys745 residue [102] were observed, all of which are in the binding active pocket of the EGFR protein (Figure 14).

In recent years, computational modeling tools for peptides have advanced significantly [108,109,110,111,112]. Similar to mAbs, peptides exhibit large and chemically diverse binding interfaces, but they show better biodistribution and tissue penetration than mAbs. Most current computational methods are based on building a model to predict whether peptides bind to antigens based on their sequences [111,113,114,115,116]. Some models were developed using ML techniques such as NetMHC [116], which trains a neural network to classify binders/non-binders using a dataset of experimental binding affinity measurements, NetMHCpan [117], which relies on neural networks and mass spectrometry (MS) data of peptides, and Abella et al.’s model [111], which uses a random forest algorithm to classify binders from non-binders and the Anchored Peptide-MHC Ensemble Generator (APE-Gen) tool [115] to model peptides–antigen complex structures. More computationally expensive molecular simulations of peptides–antigen complex structures have produced accurate binding predictions, e.g., MD (PB/GBSA) [118] and Monte Carlo simulations (available in the Rosetta modeling suite) [110].

Guardiola et al. [110] reported the exploitation of the computational power of the Rosetta modeling suite (https://www.rosettacommons.org/software (accessed on 28 February 2023)) in terms of large-scale backbone sampling (Monte Carlo simulations), side-chain composition and energy scoring to design heterochiral cyclic peptides that bind to a protein surface of PD-1. The authors designed a library of seven cyclic peptides (PD-i1- PD-i7) from those two peptides, PD-i3 (**61**) and PD-i6 (**62**). PD-i3 and PD-i6 showed mid-micromolar binding affinity to the target PD-1 and outcompeted endogenous PD-1 ligands for disrupting the PD-1/PD-L1 interaction, as shown in Figure 15.

A very important stage in the peptide de novo design developed by Guardiola et al. [110] was the hotspot selection. Two approaches were carried out, first for the peptides PD-i1 to PD-i4 and second for the peptides PD-i5 to PD-i7. These two groups of peptides were designed based on the X-ray structure of the PD-1/PD-L1 complex (PDB ID:4ZQK) and the apo-PD-1 form of the protein (PDB ID: 3RRQ), respectively [110]. The residues Tyr and Trp were selected as main hotspot residues of the interaction for the first and second approaches, using the in silico Ala scan and the FTMap algorithm (https://ftmap.bu.edu (accessed on 28 February 2023)), respectively [110].

AlphaFold, which was already mentioned in Section 3.2, is Alphabet’s DeepMind entry in the CASP13 competition as an artificial intelligence system designed to predict protein–protein or protein–peptide structures [119,120]. Recently, DeepMind and EMBL-EBI developed the AlphaFold Protein Structure Database (https://alphafold.com/ (accessed on 17 March 2023)), leading to an unprecedented number of reliable protein structure predictions that are easily accessible to the scientific community. Varadi et al. [121] provided a brief overview and highlighted how beneficial this influx of protein structure data can be for the fields of bioinformatics, structural biology and drug discovery.

### 4.2. Target Predictions of mAbs

In silico design methods typically involve molecular modeling and simulation techniques to predict the effects of point mutations on the Ab–antigen interactions. This can include methods such as MD simulations, which use classical mechanics to simulate the movement of atoms and molecules. To improve binding modes and affinity, models of mutational dynamics of the mAb variable regions can be used, considering all the complexities associated with mAb diversity, such as the natural V(D)J rearrangement process. In this regard, adapted Markov Chain Monte Carlo simulations have been used to simulate the effects of mutations on the population of Ab molecules and to predict the most likely route to the high-affinity binding [122].

## 5. Targeted Therapies for PD-1/PD-L1 ICIs

Nowadays, targeted-tailored and personalized medicine, focused on the individual characteristics of each patient’s cancer, has become an extremely topical topic. The use of computational modeling for PD-1/PD-L1 ICIs [106,107] provides a unique tool for the prediction of responses to immunotherapeutic treatments, which can be used to integrate robust data input—big data—from genomic and transcriptomic studies, clinical data and in vivo and in vitro experimental models. Jiang et al. [106] reported a computational analysis of resistance (CARE), a computational method focused on targeted therapies, to infer genome-wide transcriptomic signatures of drug efficacy from cell line compound screens. The authors predicted PRKD3 (protein kinase D3) as a potential regulator of anti-HER2 breast cancer cell resistance and validated it as a promising target to increase lapatinib (**60**), as shown in Figure 13, and trastuzumab (mAb) efficacy through combinatorial treatment with PRKD inhibitors, which can be associated with an anti-PD-1 clinical response. Furthermore, Lombardo et al. [107] carried out a similar approach in neuroblastoma, creating a computational network model simulating the different intracellular pathways involved in neuroblastoma using the COPASI software. The authors showed that ALK (anaplastic lymphoma kinase) is an important factor in PD-L1 expression since treatment with crizotinib (a known ALK inhibitor) (**61**), as shown in Figure 13, decreased PD-L1 concentrations [107].

Considering the target predictions of small molecules and peptides with important intracellular pathways is also relevant, indeed, while it is essential to understand direct bindings of new candidates to PD-1 and PD-L1, it is equally important to identify how molecules could bind and influence intracellular pathways and result in meaningful efficacy of ICIs. In the field of chemical interaction with the cell pathways, and pathway cross-talk, the PubChem Pathway [123], the successor to the legacy Pathway Interaction Database (PID), constitutes a suitable analysis entry point as it strive to connect the pharmacology to physiology. Pathway cross-talk can be assessed by bridging global databases such as KEGG Pathways [33] or Reactome [124] to any of the more focused ones, such as NetPah [125] or HumanCyc [126]. This bridging effort is being eased by the development of common formats and collaborative platforms, as it is in the case of WikiPathways [127].

## 6. Conclusions

We would like to emphasize the need for computational approaches to immune oncology therapies. We focused on using data-driven methods to identify potential targets for PD-1/PD-L1 ICIs and to develop novel drug candidates such as antibodies, peptides or small molecules. We highlighted the use of cheminformatics and bioinformatics tools to analyze large datasets of molecules, gene expression and protein–protein interaction data to identify potential biomarkers that can be used to predict response to PD-1/PD-L1 inhibitors. We also discussed the use of computer-aided drug design structure- and ligand-based virtual screening processes, molecular docking, homology modeling and MD simulation methodologies essential for successful drug discovery campaigns focusing on small molecule ICIs. We compiled a list of databases and web tools used in the context of cancer and immunotherapy, which is available as Appendix A for this review. This includes databases and tools of a general scope, cancer and immunology.

## Figures and Tables

**Figure 1 ijms-24-05908-f001:**
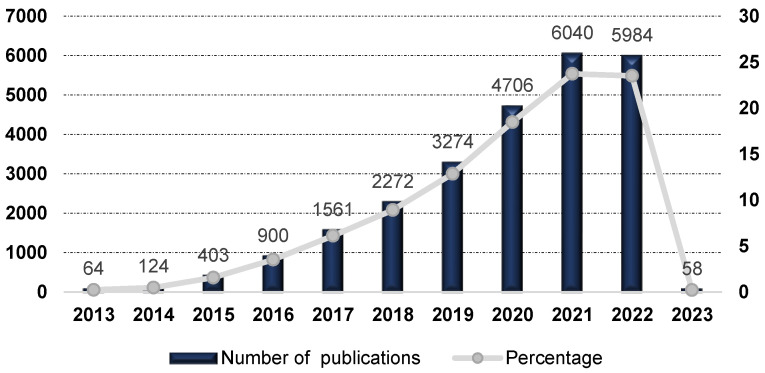
Number of publications and percentages per year covering ICIs in cancer topics, period 2013–2023. Data source from Web of Science™ Core Collection.

**Figure 2 ijms-24-05908-f002:**
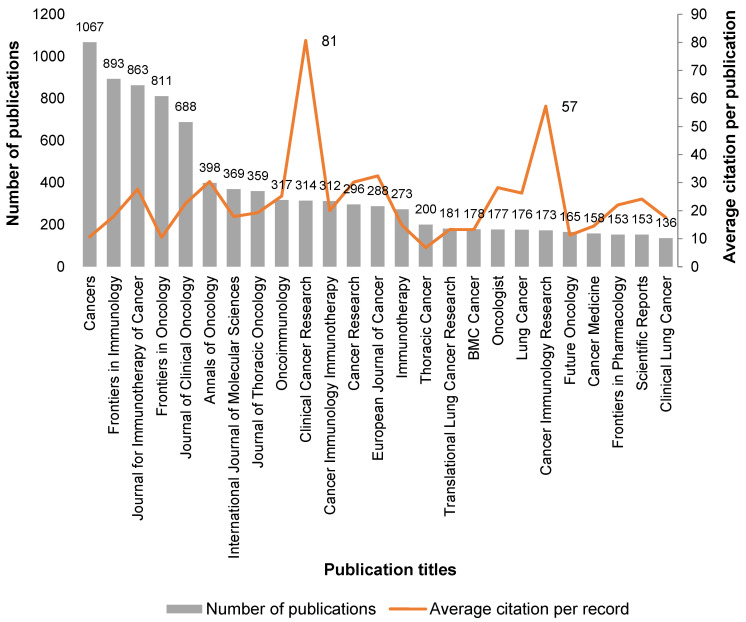
Analysis of the 25 selected journals reporting ICIs in cancer topics, by number of records and average citation per record, since 2013. Data source from Web of Science™ Core Collection.

**Figure 3 ijms-24-05908-f003:**
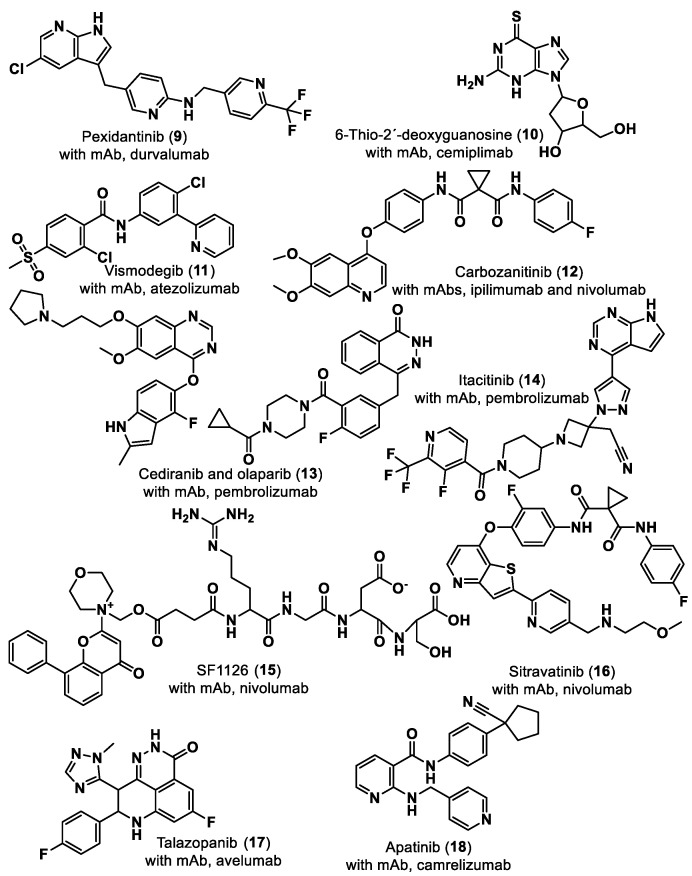
Small molecule (**9**–**18**) ICIs targeting PD-1/PD-L1 in combination clinical trials with mAbs.

**Figure 4 ijms-24-05908-f004:**
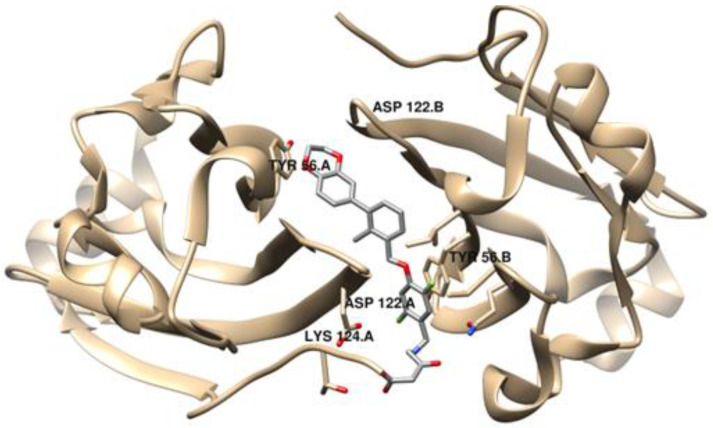
Visualization of the co-crystal dimer structure of PD-L1 in complex with BMS-200 (Protein Data Bank, PDB ID: 5N2F), highlighted with the critical residues for ligand binding, using UCSF Chimera [57,58].

**Figure 5 ijms-24-05908-f005:**
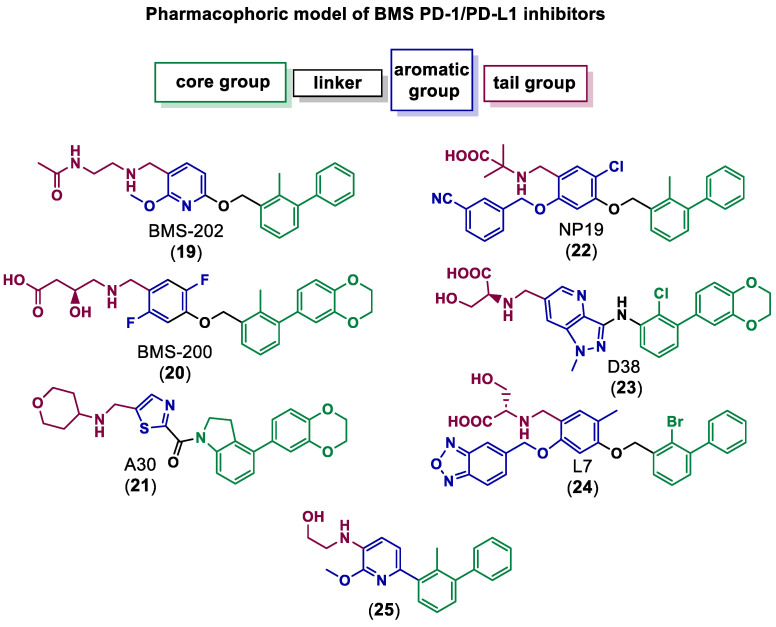
Small molecule inhibitors of PD-1/PD-L1 based on the BMS pharmacophoric model [59,61,62,74,75,76].

**Figure 6 ijms-24-05908-f006:**
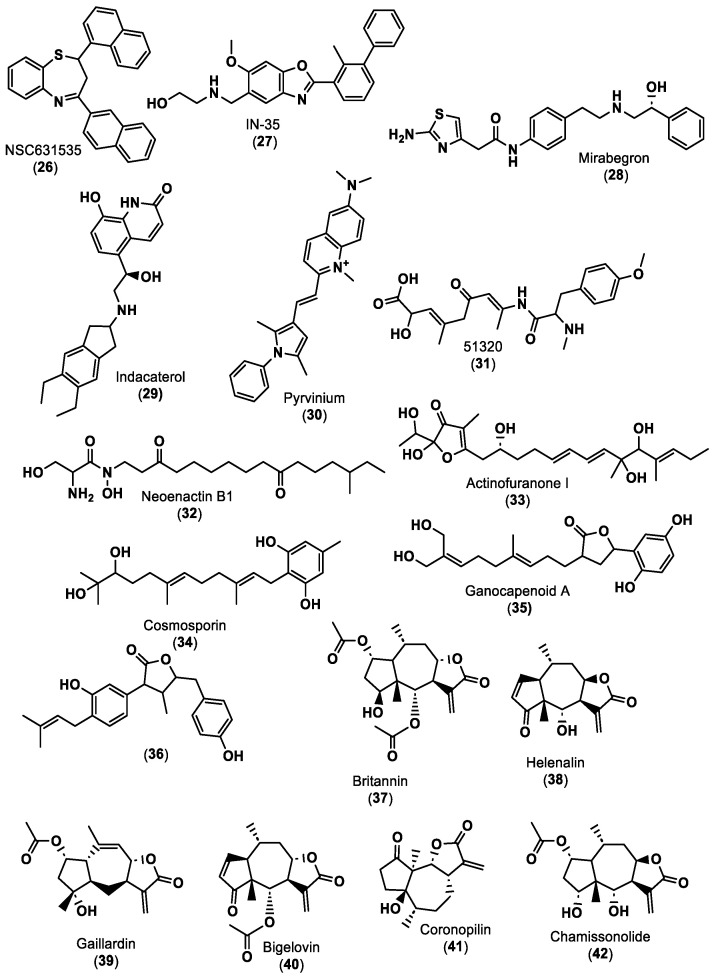
Small molecule inhibitors of PD-1/PD-L1 immune checkpoint [51,65,77,78,80,81,82].

**Figure 7 ijms-24-05908-f007:**
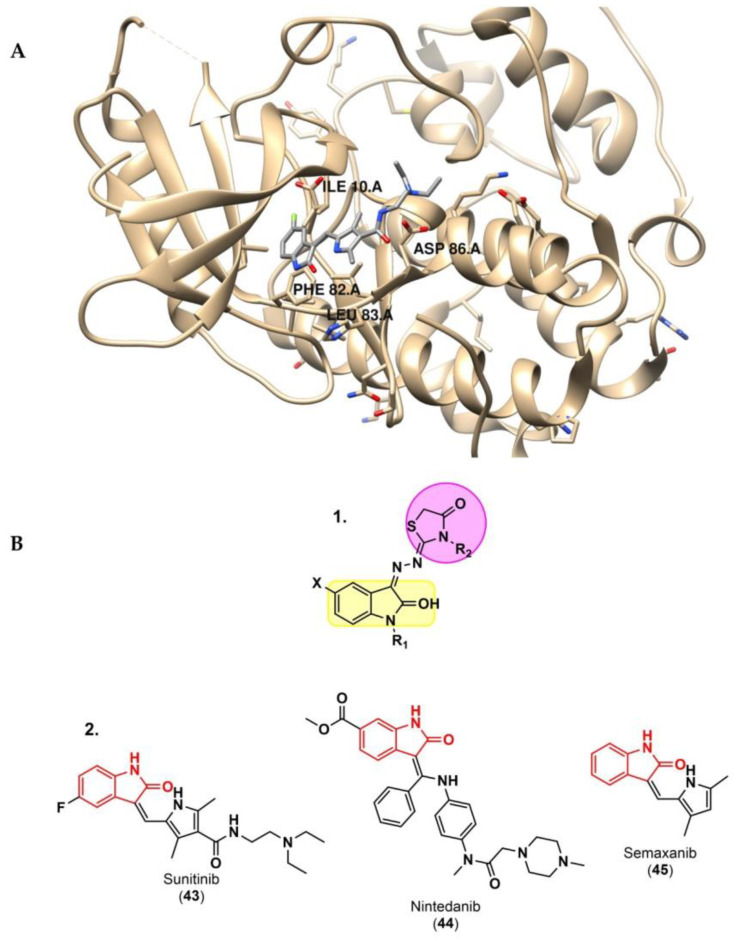
(**A**) 3D interactions of sunitinib with CDK2 (PDB code: 3Ti1) using UCSF Chimera [58,84]. (**B**) 1. Structure of the designed hybridized molecules. 2. 2-indolinone pharmacophore motifs in CDK2 inhibitor commercially marketed drugs [58,84].

**Figure 8 ijms-24-05908-f008:**
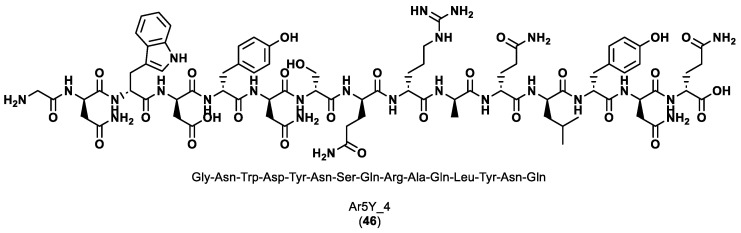
Sequence and structure of the most potent peptide designed, Ar5Y_4 [86].

**Figure 9 ijms-24-05908-f009:**
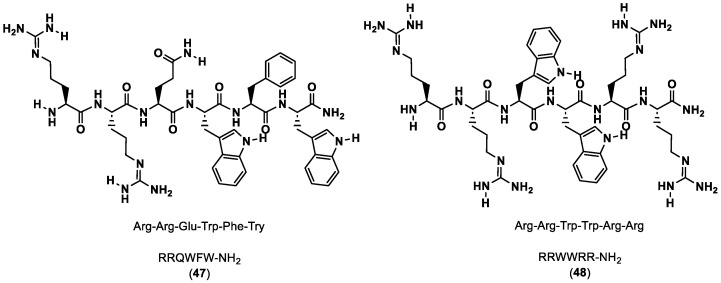
Sequence and structure of RRQWFW-NH_2_ and RRWWRR-NH_2_ [90].

**Figure 10 ijms-24-05908-f010:**
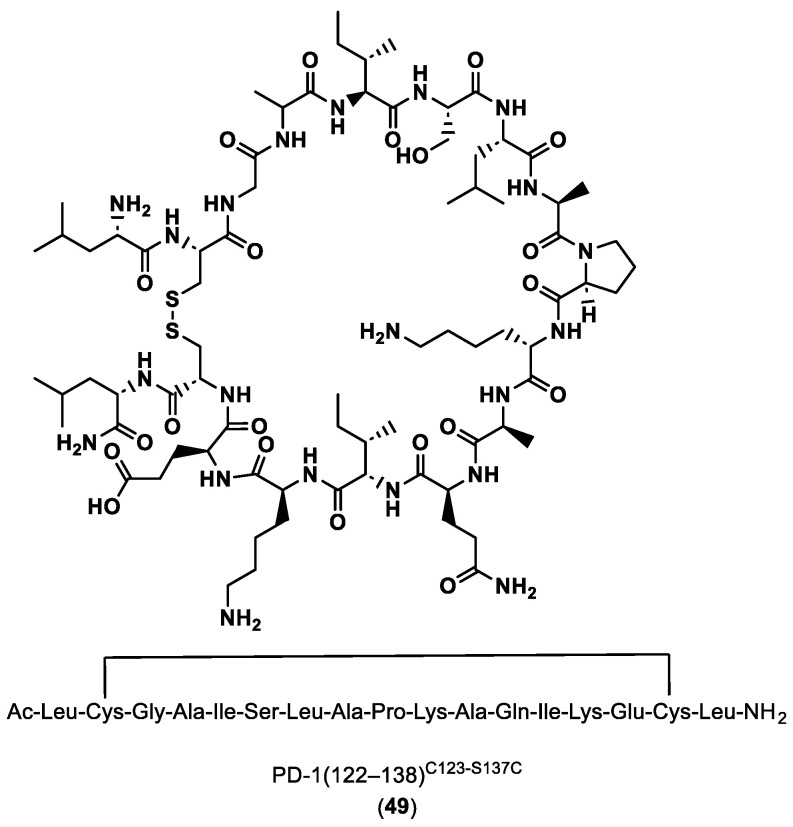
Sequence and structure of PD-1(122–138)^C123-S137C^ [91].

**Figure 11 ijms-24-05908-f011:**
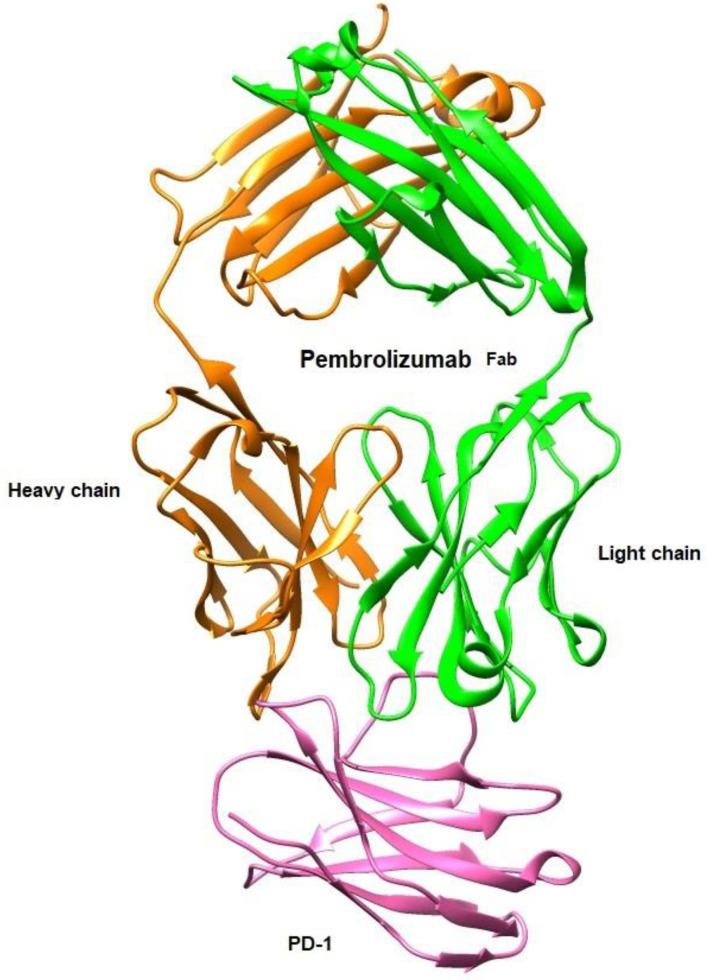
Structural representation of the antigen-binding fragment (Fab) of pembrolizumab (PDB ID: 5GGS) in complex with the extracellular domain of human PD-1 receptor using UCSF Chimera [58,97,98].

**Figure 12 ijms-24-05908-f012:**
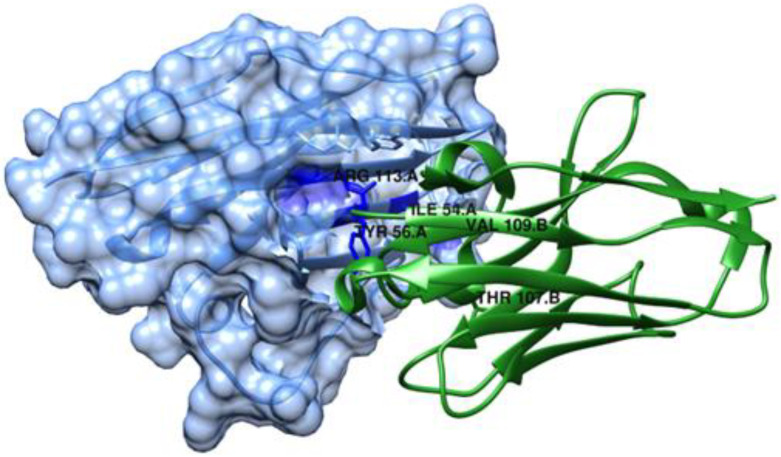
Structure of the KN035/ PD-L1 complex (PDB ID: 5JDS) represented using UCSF Chimera [58,99]. PD-L1 is shown as a slate, semi-transparent surface.

**Figure 13 ijms-24-05908-f013:**
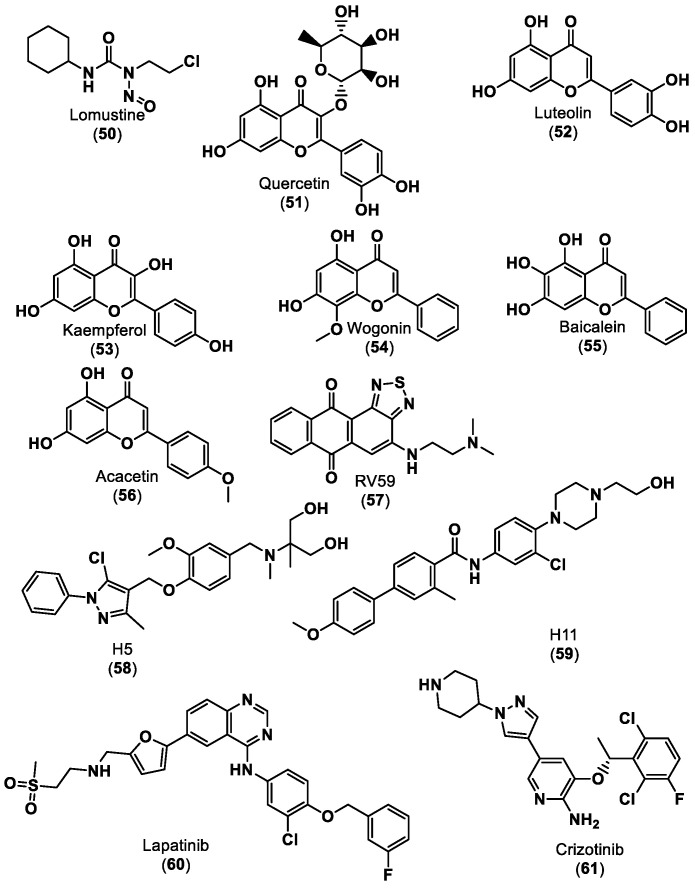
Small molecule ICIs predict targets that regulate PD-L1 expression [101,102,103,106,107].

**Figure 14 ijms-24-05908-f014:**
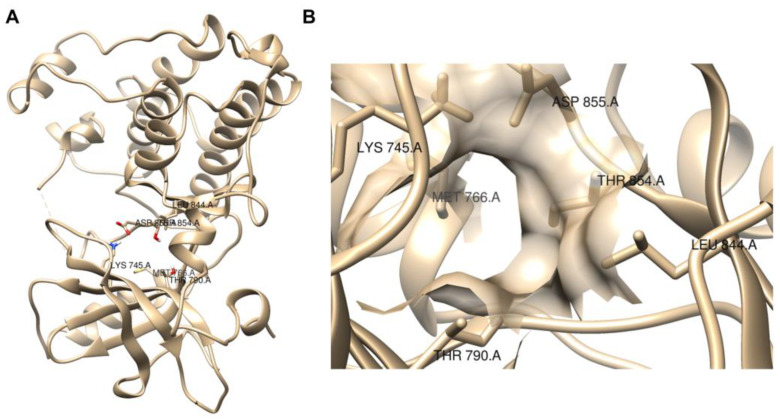
Visualization of EGFR protein (PDB, ID 3W2S) with UCSF Chimera [58]. (**A**) Full view; (**B**) binding active pocket region [101,102].

**Figure 15 ijms-24-05908-f015:**
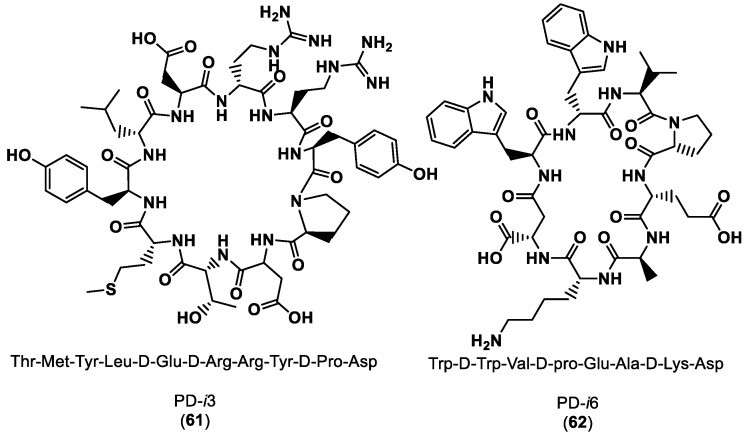
Structures and sequences of the designed peptide inhibitors of PD-1 [110].

**Table 1 ijms-24-05908-t001:** Small molecule (**1–8**) ICIs targeting PD-1/PD-L1 in clinical trials.

Target [8]	Compound [8]	Chemical Structure	Clinical State [8]	Indications [8]
PD-1/PD-L1	CA-170 (**1**)	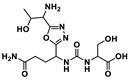	Phase 1	Advanced solid tumors or lymphomas
PD-1/PD-L1	Tomivosertib (**2**)	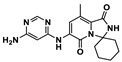	Phase 2	Solid tumors
PD-1/PD-L1	Navtemadlin (**3**)	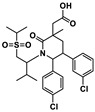	Phase 1	Merkel cell carcinoma
PD-1/PD-L1	Abemaciclib (**4**)	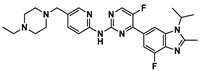	Phase 2	Head and neck neoplasms
PD-L1	INCB086550 (**5**)	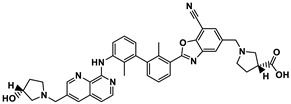	Phase 2	Solid tumors
PD-L1	Ciforadenant (**6**)	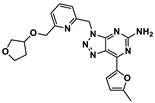	Phase 1	Renal cell cancer
PD-L1	ABSK043 (**7**)	--- ^1^	Phase 1	Neoplasms
PD-L1	ASC61 (**8**)	--- ^1^	Phase 1	Advanced solid tumors

^1^ Indicates structural uncertainty.

## Data Availability

The databases and web tools reported in this review are available as Appendix A.

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
