# Peer review of "Computational Approaches Drive Developments in Immune-Oncology Therapies for PD-1/PD-L1 Immune Checkpoint Inhibitors"

_ijms, 2023, doi:10.3390/ijms24065908_

Round 1

Reviewer 1 Report

The review by Sobral and colleagues aims to extensively summarize the computational approaches used by other teams in the development of immune checkpoint inhibitor candidates targeting the PD-1/PD-L1 interaction.

In particular, this work describes in detail the development steps for small molecules, peptides and antibodies targeting the PD-1 pathway, with the support of well-chosen references.

But this work goes beyond the specificity of PD-1, and thus the description of the methods employed, as well as the exhaustive accumulation of these in supplementary material, will definitely help and inspire other teams for other therapeutic targets.

It is a very interesting work, meticulous, pleasant to read and highly valuable for the scientific community. Congratulations and thank you to the authors!

Author Response

Manuscript number: ijms-2282403
Title: " Computational approaches drive developments in immune-oncology
therapies for PD-1/PD-L1 immune checkpoint inhibitors" 

Response to Reviewers

We would like to thank Reviewers 1 and 2, and the Editor for their comments, concerns, and revisions, which contributed to improving the quality of the revised manuscript. The manuscript was revised according to all of Reviewer 2's comments. Answers and alterations in the revised manuscript can be found using the "Track Changes" function and are highlighted in yellow for easy identification.

Reviewer#2
Critical I
In section 1.2. Publication Trends from 2013 to 2022 on ICIs, the authors rank journals based on the number of studies that have been published regarding ICIs. Despite being informative on the global interest that had had this topic in science in the last 10 years, I would like to see also a rank based on the top cited articles. So, I would like to see a combined metric that takes into account also the citations that these articles have obtained and then ranked journals based on it.
Answer
We are grateful to Reviewer 2 for raising this issue, which allowed us to improve the manuscript. Figure 2 was changed to take into account the citation of the top 25 journals. An analysis of the citation profile was added to the revised version of the manuscript in Section 1.2.
Manuscript
Although we perceive that, in general, older publications accumulate more citations, and the publications analyzed in this theme are mostly very recent (representing 94% in the period 2017-2022), there are clearly two journals that stand out with a number of citations per publication of 81 and 57 for Clinical Cancer Research and Cancer Immunology Research, respectively (see Figure 2). Interestingly, these two most-cited journals are both from the American Association for Cancer Research publisher.
Critical II
One of the main interests of the authors in the review is to focus on docking predictions and molecular dynamics. In this field, Alphafold has been a life-changing paradigm. I believe that it would be beneficial to include also a paragraph on how Alphafold can improve peptide predictions for immune checkpoint inhibitors. A possible starting point for the authors could be the following study: https://www.jbc.org/article/S0021-9258(22)01212-1/fulltext
Answer
We are grateful to Reviewer 2 for raising this issue, which allowed us to improve the manuscript. Two new paragraphs were added to Sections 3.2 and 4.1, highlighting the AlphaFold tool. The AlphaFold Protein Structure Database was also added to the Supplementary Materials of the manuscript.
Manuscript
Section 3.2
One of the most recently developed computational methods, AlphaFold, has revolutionized structural biology by predicting highly accurate structures of proteins and their complexes with peptides, antibodies, and proteins [92-94]. Besides that, AlphaFold can also be useful for protein-peptide systems and drug discovery, as it can help identify the highest affinity binder among a set of peptides. Chang & Perez [92] presented a new competitive binding assay using AlphaFold to predict the structures of the receptor in the presence of peptides. The authors tested the application on six protein receptors for which they possessed the experimental binding affinities to several peptides, and obtained the predicted structures (bound form) for the higher affinity peptides [92]. They concluded that this assay had a better prediction score for identifying stronger peptide binders that also adopt stable secondary structures upon binding [92]. Johansson-Åkhe et al.[93] also show the importance of AlphaFold, in this case AlphaFold-Multimer. AlphaFold-Multimer is shown to be capable of predicting the structure of peptide-protein complexes with acceptable or better accuracy than previous models, and also has the capacity to predict whether a peptide and a protein will interact, thus improving peptide-protein docking [93].
Section 4.1
AlphaFold, which was already mentioned in Section 3.2, is Alphabet's DeepMind entry in the CASP13 competition as an artificial intelligence system designed to predict protein-protein or protein-peptide structures [119,120]. Recently, DeepMind and EMBL-EBI developed the AlphaFold Protein Structure Database (https://alphafold.com/), leading to an unprecedented number of reliable protein structure predictions that are easily accessible to the scientific community. Varadi et al. [121] provided a brief overview and highlighted how beneficial this influx of protein structure data can be for the fields of bioinformatics, structural biology, and drug discovery.
References
92.    Chang, L.; Perez, A. Ranking Peptide Binders by Affinity with AlphaFold. Angew. Chem., Int. Ed. Engl. 2023, 62, e202213362, doi:10.1002/anie.202213362.
93.    Johansson-Akhe, I.; Wallner, B. Improving peptide-protein docking with AlphaFold-Multimer using forced sampling. Front. bioinform. 2022, 2, 959160, doi:10.3389/fbinf.2022.959160.
94.    Jumper, J.; Evans, R.; Pritzel, A.; Green, T.; Figurnov, M.; Ronneberger, O.; Tunyasuvunakool, K.; Bates, R.; Zidek, A.; Potapenko, A.; Bridgland, A.; Meyer, C; Kohl, S.A.A.; Ballard, A.J.; Cowie, A.; Romera-Paredes, B.; Nikolov, S.; Jain, R.; Adler, J.; Back, T.; Petersen, S.; Reiman, D.; Clancy, E; Zielinski, M.; Steinegger, M.; Pacholska, M.; Berghammer, T.; Bodenstein, S.; Silver, D.; Vinyals, O.; Senior, A.W.; Kavukcuoglu, K.; Kohli, P.; Hassabis, D. Highly accurate protein structure prediction with AlphaFold. Nature 2021, 596, 583-589, doi:10.1038/s41586-021-03819-2.
119.    Marcu, S.B.; Tabirca, S.; Tangney, M. An Overview of Alphafold's Breakthrough. Front. Artif. Intell. 2022, 5, 875587, doi:10.3389/frai.2022.875587.
120.    AlQuraishi, M. AlphaFold at CASP13. Bioinformatics 2019, 35, 4862-4865, doi:10.1093/bioinformatics/btz422.
121.    Varadi, M.; Velankar, S. The impact of AlphaFold Protein Structure Database on the fields of life sciences. Proteomics 2022, e2200128, doi:10.1002/pmic.202200128.
Critical III
Because the aim of the authors is to provide a comprehensive review on the state of the art of computational models applied in PD1/PDL1 axis, I believe that they should include also a paragraph regarding pathway analysis. Despite the authors touching briefly on this topic in section 4.1 Target predictions of small molecules and peptides, mentioning the cross-talk with other important pathways such as AKT and EGFR, I believe that this topic should be a subparagraph on its own. Indeed, while it is important to understand chemical bindings between molecules (antibodies, peptides, small molecules…) and their targets (i.e. PD1), it is equally important to understand whether this binding would reflect a meaningful change in the phenotype. For doing so, several studies have proposed pathways analysis for predicting the efficacy of PD1 immunotherapy, as shown in the context of Neuroblastoma (https://pubmed.ncbi.nlm.nih.gov/31480495/).  
Answer
We thank the reviewer 2 suggestion, and we have included a new Section 5 dedicated to computational methods focused on targeted therapies and we have changed Figure 13.
Manuscript
5. Targeted therapies for PD-1/PD-L1 ICIs
Nowadays, targeted-tailored and personalized medicine, focused on the individual characteristics of each patient's cancer, has become an extremely topical topic. The use of computational modeling for PD-1/PD-L1 ICIs [106,107] provides a unique tool for the prediction of responses to immunotherapeutic treatments, which can be used to integrate robust data input - big data -  from genomic and transcriptomic studies, clinical data, and in vivo and in vitro experimental models. Jiang et al. [106] reported a computational analysis of resistance (CARE), a computational method focused on targeted therapies, to infer genome-wide transcriptomic signatures of drug efficacy from cell line compound screens. The authors predicted PRKD3 (protein kinase D3) as a potential regulator of anti-HER2 breast cancer cells resistance and validated it as a promising target to increase lapatinib (60), as shown in Figure 13, and trastuzumab (mAb) efficacy through combinatorial treatment with PRKD inhibitors, which can be associated with an anti-PD-1 clinical response. Furthermore, Lombardo et al. [107] carried out a similar approach in neuroblastoma, creating a computational network model simulating the different intracellular pathways involved in neuroblastoma using the COPASI software. The authors showed that ALK (anaplastic lymphoma kinase) is an important factor in PD-L1 expression since treatment with crizotinib (a known ALK inhibitor) (61), as shown in Figure 13, decreased PD-L1 concentrations [107].
Considering the target predictions of small molecules and peptides with important intracellular pathways is also relevant. Indeed, while it is essential to understand direct bindings of new candidates to PD-1 and PD-L1, it is equally important to identify how molecules could bind and influence intracellular pathways and result in meaningful efficacy of ICI. On the field of chemical interaction with the cell pathways, and pathway cross-talk, the PubChem Pathway [123], the successor to the legacy Pathway Interaction Database (PID), constitutes a suitable analysis entry point as it strive to connect the pharmacology to physiology. Pathway cross-talk can be assessed by bridging global databases like KEGG Pathways [33] or Reactome [124] to any of the more focused ones as NetPah [125] or HumanCyc [126]. This bridging effort is being eased by the development of common formats, and collaborative platforms as it is the case of WikiPathways [127].
Reference
107.    Lombardo, S.D.; Presti, M.; Mangano, K.; Petralia, M.C.; Basile, M.S.; Libra, M.; Candido, S.; Fagone, P.; Mazzon, E.; Nicoletti, F.; Bramanti, A. Prediction of PD-L1 Expression in Neuroblastoma via Computational Modeling. Brain Sci. 2019, 9, 221, doi:10.3390/brainsci9090221.
123.    Kim, S.; Cheng, T.; He, S.; Thiessen, P.A.; Li, Q.; Gindulyte, A.; Bolton, E.E. PubChem Protein, Gene, Pathway, and Taxonomy Data Collections: Bridging Biology and Chemistry through Target- Centric Views of PubChem Data. J. Mol. Biol. 2022, 434, doi:10.1016/j.jmb.2022.167514.
124.    Gillespie, M.; Jassal, B.; Stephan, R.; Milacic, M.; Rothfels, K.; Senff-Ribeiro, A.; Griss, J.; Sevilla, C.; Matthews, L.; Gong, C.; et al. The reactome pathway knowledgebase 2022. Nucleic Acids Res. 2022, 50, D687-D692, doi:10.1093/nar/gkab1028.
125.    Kandasamy, K.; Mohan, S.S.; Raju, R.; Keerthikumar, S.; Kumar, G.S.S.; Venugopal, A.K.; Telikicherla, D.; Navarro, J.D.; Mathivanan, S.; Pecquet, C.; et al. NetPath: a public resource of curated signal transduction pathways. Genome Biol. 2010, 11, doi:10.1186/gb-2010-11-1-r3.
126.    Caspi, R.; Altman, T.; Billington, R.; Dreher, K.; Foerster, H.; Fulcher, C.A.; Holland, T.A.; Keseler, I.M.; Kothari, A.; Kubo, A.; et al. The MetaCyc database of metabolic pathways and enzymes and the BioCyc collection of Pathway/Genome Databases. Nucleic Acids Res. 2014, 42, D459-D471, doi:10.1093/nar/gkt1103.
127.    Martens, M.; Ammar, A.; Riutta, A.; Waagmeester, A.; Slenter, D.N.; Hanspers, K.; Miller, R.A.; Digles, D.; Lopes, E.N.; Ehrhart, F.; et al. WikiPathways: connecting communities. Nucleic Acids Res. 2021, 49, D613-D621, doi:10.1093/nar/gkaa1024.

Reviewer 2 Report

Sobral et al. provide a detailed review of the computational approaches that have been used in the last five years to ameliorate the efficacy of drugs targeting the PD1/PDL1 axis in cancer. This topic is very relevant based on the impact that can have on the clinic and the rising number of studies on this subject.

In section 1.2. Publication Trends from 2013 to 2022 on ICIs, the authors rank journals based on the number of studies that have been published regarding ICIs. Despite being informative on the global interest that had had this topic in science in the last 10 years, I would like to see also a rank based on the top cited articles. So, I would like to see a combined metric that takes into account also the citations that these articles have obtained and then ranked journals based on it.

One of the main interests of the authors in the review is to focus on docking predictions and molecular dynamics. In this field, Alphafold has been a life-changing paradigm. I believe that it would be beneficial to include also a paragraph on how Alphafold can improve peptide predictions for immune checkpoint inhibitors. A possible starting point for the authors could be the following study:

https://www.jbc.org/article/S0021-9258(22)01212-1/fulltext

Because the aim of the authors is to provide a comprehensive review on the state of the art of computational models applied in PD1/PDL1 axis, I believe that they should include also a paragraph regarding pathway analysis. Despite the authors touching briefly on this topic in section 4.1 Target predictions of small molecules and peptides, mentioning the cross-talk with other important pathways such as AKT and EGFR, I believe that this topic should be a subparagraph on its own. Indeed, while it is important to understand chemical bindings between molecules (antibodies, peptides, small molecules…) and their targets (i.e. PD1), it is equally important to understand whether this binding would reflect a meaningful change in the phenotype. For doing so, several studies have proposed pathways analysis for predicting the efficacy of PD1 immunotherapy, as shown in the context of Neuroblastoma (https://pubmed.ncbi.nlm.nih.gov/31480495/).  

Author Response

(The authors gave the same response as above.)
